# BDNF as a Mediator of Antidepressant Response: Recent Advances and Lifestyle Interactions

**DOI:** 10.3390/ijms232214445

**Published:** 2022-11-21

**Authors:** Susana Cubillos, Olivia Engmann, Anna Brancato

**Affiliations:** 1Institute for Biochemistry and Biophysics, Friedrich-Schiller-University Jena, 07745 Jena, Germany; 2Department of Sciences for Health Promotion and Mother and Child Care “G. D’Alessandro”, University of Palermo, 90127 Palermo, Italy

**Keywords:** antidepressants, depression, BDNF, TrkB, environment

## Abstract

Conventional antidepressants are widely employed in several psychiatric and neurologic disorders, yet the mechanisms underlying their delayed and partial therapeutic effects are only gradually being understood. This narrative review provides an up-to-date overview of the interplay between antidepressant treatment and Brain-Derived Neurotrophic Factor (BDNF) signaling. In addition, the impact of nutritional, environmental and physiological factors on BDNF and the antidepressant response is outlined. This review underlines the necessity to include information on lifestyle choices in testing and developing antidepressant treatments in the future.

## 1. The Paradigm Shift in Antidepressant Mechanism of Action

Antidepressants are widely used drugs indicated in several psychiatric and neurologic disorders, including major depressive disorder (MDD), anxiety, obsessive compulsive disorder and neuropathic pain. These drugs facilitate the signaling of serotonin and/or norepinephrine either by blocking their reuptake to presynaptic terminals, by inhibiting their catabolism or by binding to monoamine autoreceptors. Increased monoamine synaptic availability occurs soon after drug administration. However, the clinical antidepressant effect is delayed and typically requires several weeks of chronic treatment before an antidepressant response is obtained [1]. Thus, the enhancement of extracellular monoamine levels alone cannot explain the antidepressant effect [2]. In addition, a recent review on the link between serotonin and depression found no convincing evidence that MDD is associated with, or is caused by, low serotonin concentrations or abnormal serotonin system activity [3]. Instead, starting from the work of Duman and collaborators [4], novel molecular and cellular mechanisms were identified, highlighting the role of second messenger systems, which are involved in gene expression regulation [5]. In detail, the signaling of the neurotrophin Brain-Derived Neurotrophic Factor (BDNF), activated by the second messenger cyclic AMP system, accounts for adaptive changes in intracellular signal transduction and synaptic connectivity, which emerged as the relevant mechanism of action of antidepressant medications [6]. The delayed therapeutic effect of chronic antidepressant treatment has been traditionally attributed to the progressive neuroplastic changes involving monoaminergic transmissions, including the progressive desensitization of upregulated serotonin receptors [7]. However, progressive neuronal plasticity resulting from the recruitment of BDNF signaling, which is indicated as central to the antidepressant response, is also consistent with the delayed therapeutic effect observed after chronic antidepressant treatment [8]. As a game changer in this scenario, the dissociative anesthetic and N-methyl-D-aspartate (NMDA) receptor inhibitor ketamine showed rapid and sustained antidepressant action in clinical trials [9] and the (S)-ketamine nasal spray has been developed and approved in the United States and Europe for the employment in treatment-resistant depression [10]. Interestingly, it becomes increasingly clear that antidepressants with different molecular targets, including monoaminergic drugs and ketamine share BDNF-mediated neuroanatomical effects [11,12].

BDNF binds with a high affinity to the tropomyosin receptor kinase B (TrkB), a tyrosine kinase receptor expressed both pre- and post-synaptically [13]. The binding of BDNF to TrkB induces the rapid receptor autophosphorylation, which promotes at least three intracellular cascades [14]. Remarkably, TrkB activation is able to modulate BDNF expression levels in neurons, forming a transcriptional positive feedback loop, involving CREB family transcription factors as the main regulators of *Bdnf* gene expression after TrkB signaling [15].

TrkB is crucial to antidepressant responses. When direct BDNF infusions were administered in rodents, the inhibition of TrkB signaling prevented the antidepressant-like effects of BDNF [16,17]. Antidepressants have been shown to promote TrkB autophosphorylation and downstream CREB-signaling after both acute and repeated administration as soon as 30 min after antidepressant administration. However, this effect vanished after at 6 or 24 h [17], supporting the need for a chronic regimen. Interestingly, recent findings demonstrated that antidepressants activate TrkB independently of BDNF binding as well. The tricyclic antidepressant imipramine readily induced the phosphorylation of TrkB in conditional *Bdnf*^−/−^ knock-out mice, indicating that BDNF is not required for TrkB activation. Moreover, TrkB phosphorylation induced by either fluoxetine, citalopram, and reboxetine was also observed in serotonin transporter-deficient mice, suggesting that antidepressant drugs transactivate TrkB independently of BDNF and monoamine transporter blockade [18,19].

Overall, the activation of TrkB triggers activity-dependent synaptic plasticity [20]. Thus, it has been postulated that antidepressant-induced TrkB signaling reactivates a state of juvenile-like plasticity in the adult brain, restructuring neural circuits and, in consequence, mood and behavior [8,21]. In particular, the chronic administration of antidepressants is able to stimulate adult hippocampal neurogenesis, interacting with TrkB on the neural progenitor cells of the dentate gyrus, inducing a continuous proliferation and neuronal differentiation, which is the source of such functional plasticity [22]. In addition, the deletion of TrkB in neural stem/progenitor cells, or the pharmacological inhibition of ERK signaling also abolished ketamine-induced behavioral responses in depression- and anxiety-like paradigms [23], suggesting that dysfunctional TrkB in the neurogenic niche can be an etiological factor for refractory depression. Experimental data are also corroborated by consistent postmortem findings of decreased prefrontal and hippocampal TrkB signaling in MDD patients who committed suicide [24,25,26,27].

The effects of monoaminergic antidepressants and ketamine on TrkB signaling have long been considered indirect, caused by the inhibition of monoamine transporters and NMDA receptors, respectively, and mediated by the increase in BDNF expression and release [19]. However, early evidence and more recent data indicate that antidepressants directly bind to TrkB, [28,29]. The work of the Castrén group elegantly showed that antidepressants of various classes directly bind to a site formed by a dimer of TrkB transmembrane domains, thereby facilitating cell surface expression of the receptor, increasing the sensitivity of the dimerized receptor for BDNF, and promoting BDNF-mediated TrkB signaling [29].

Although the affinity of antidepressants for TrkB is lower than those for monoaminergic transporters, under chronic regimen they accumulate in the brain, and the concentration achieved upon long-term use allows for direct binding to TrkB [29]. As recently highlighted by a proximity ligation assay, acute imipramine administration increased the physical interaction between BDNF and TrkB in the rat cingulate cortex 3 h later, although this effect was not long-lasting. In contrast, repeated imipramine administration exerted a durable effect, which decreased slowly and vanished only after 21 drug-free days [30].

In addition, recent evidence reveals a physical interaction between TrkB and 5-HT_2A_ receptors, in a heterodimeric complex which prevents TrkB activation in brain regions including the hippocampus, prefrontal cortex and striatum [7]. The monoaminergic antidepressant-induced neuronal events, including the progressive 5-HT receptor downregulation, thus decreased functional inhibition of TrkB, and the intracellular events triggered by direct TrkB modulation, are in accordance with the slow development of the clinical antidepressant response.

Overall, TrkB is emerging as the sly target for antidepressants (Figure 1). As such, the paradigm of the antidepressant mechanism of action shifts from the modulation of monoamine signaling to the direct promotion of BDNF transmission. Thus, alterations in upstream or downstream components of BDNF signaling can undermine the response to antidepressants. On the other hand, environmental factors which modulate BDNF signaling, or activate its downstream pathways, may provide additional tools for amplifying the antidepressant response.

## 2. BDNF as a Critical Antidepressant Mediator

### 2.1. BDNF Exerts Distinct Effects in MDD-Linked Brain Regions

BDNF is a crucial regulator of neurite outgrowth, synaptic plasticity, and the selection of functional neuronal connections in the central nervous system [20,31,32], that mediates the plastic changes induced by antidepressants [6].

The first studies unveiling the involvement of BDNF in antidepressant response showed that chronic administration of conventional antidepressant drugs, including tranylcypromine, sertraline, desipramine, and mianserin, significantly increased BDNF mRNA expression in the hippocampus and cortical regions, counteracting the downregulation of *Bdnf* mRNA in the hippocampus observed in response to restraint stress [4,33].

When heterozygous *Bdnf* null (*Bdnf*^+/−^) mice were employed to assess BDNF’s role in the antidepressant response, a ~50% reduction in BDNF levels did not impact depression-related behavior per se, but the antidepressant imipramine was ineffective in the forced swim test [17]. In addition, when BDNF was selectively deleted in the forebrain of inducible knockout mice, no basal alterations in depression-related behavior were highlighted. Still, an attenuated response to the tricyclic antidepressant desipramine was observed [34], further indicating that forebrain BDNF was required for antidepressant efficacy.

To directly examine the causal involvement of BDNF in antidepressant responses, the BDNF protein was infused directly into the midbrain, and an antidepressant-like effect was observed in two animal models of MDD [35]. Subsequent work showed that a single bilateral infusion of a low dose of BDNF into the dentate gyrus or CA3, but not CA1, region of the hippocampus was sufficient to exert an antidepressant-like effect in forced swim test and learned helplessness paradigms, without affecting locomotor activity or passive avoidance [36]. This suggested that BDNF may act as a region-specific effector of the antidepressant action. Notably, the effects of the hippocampal BDNF infusion were rapid (within three days), long-lasting (at least ten days) and recapitulated the therapeutic effects of repeated imipramine treatment [36]. In addition, the involvement of hippocampal BDNF, specifically in the dentate gyrus, in antidepressant efficacy was demonstrated by the employment of a viral-mediated gene transfer approach, which allowed for selectively deleting BDNF in discrete hippocampal subregions. The Cre-dependent deletion of BDNF in the dentate gyrus or CA1 subregion of the hippocampus in floxed *Bdnf* mice was not sufficient to induce depression-like behavior. However, the loss of BDNF in the dentate gyrus, but not the CA1, was crucial for the antidepressant-like effect of conventional antidepressants such as citalopram and desipramine in the forced swim test [37].

Although the role of BDNF in antidepressant effects has been consistently shown, the mechanisms through which antidepressants activate BDNF signaling remain unclear. BDNF is anterogradely transported to nerve terminals and released in response to depolarization [38,39]. Antidepressant-induced increases in synaptic availability of serotonin and norepinephrine levels were suggested to locally promote BDNF release [40,41]. In detail, incubation of cultured raphe neurons with serotonin increased BDNF release, which in turn enhanced serotonergic differentiation of the neurons [41]. Moreover, both preclinical and clinical research show a close relationship between antidepressants and alterations in *Bdnf* expression, indicating that the *Bdnf* gene is a critical epigenetic target [42].

Nevertheless, the role of BDNF in depression pathogenesis is not univocal. Rather, it depends on the brain region and individual circuits. As such, while BDNF exerts an antidepressant effect in the prefrontal cortex [43] and hippocampus [36], an opposite role has been highlighted in the mesolimbic circuit, including dopamine projections from the ventral tegmental area to the nucleus accumbens [44,45]. Interestingly, in situ hybridization of *Bdnf* mRNA highlighted that chronic fluoxetine administration increased *Bdnf* expression not only in hippocampal cell bodies, but also in the mesocorticolimbic circuit, including ventral tegmental area, prefrontal cortex, and nucleus accumbens shell, whereas no changes were detected in the substantia nigra and dorsal striatum [46]. This study did not report behavioral correlates of antidepressant response, but a strong BDNF activity in the mesocorticolimbic pathway may be a source of treatment resistance.

Indeed, the inhibition of BDNF activity in the mesolimbic pathway exerts antidepressant-like effects in rodents and reverses the social avoidance behavior caused by chronic social defeat stress paradigm [47,48]. In addition, a recent study by Furuse and colleagues (2019) further implicates the changes in BDNF levels in the nucleus accumbens in the pathophysiology and treatment of treatment-resistant depression [49]. Specifically, the antidepressant response to escitalopram in the resistant-depression model, which combines the synergic pro-depressant effects of early life adversity, such as prenatal ethanol exposure [50], and adolescent corticosterone exposure, was associated with decreased BDNF levels in the serum and the nucleus accumbens of rats.

### 2.2. Multiple Cell Types Contribute to Increased BDNF Following Antidepressant Treatment

While significant work has implicated hippocampal and cortical BDNF in antidepressant responses, only a few reports looked at the contribution of specific cell types in the brain, which produce BDNF following antidepressant treatment. In this regard, BDNF is predominantly expressed by neurons and it is conceivable that cortical and hippocampal interneurons are involved in the actions of antidepressants, contributing to the storage, production, and release of BDNF [51,52].

However, the antidepressant-induced increase in BDNF levels might also result from the upregulation in neurons targeting the prefrontal and hippocampal regions. For instance, the selective ablation of the BDNF-promoting transcription factor cAMP response element binding protein (CREB) in serotonergic neurons counteracted the upregulation of BDNF in the hippocampus and prefrontal cortex, evoked by chronic fluoxetine administration [53]. In addition, the selective BDNF overexpression in serotonergic terminals, effective in inducing antidepressant-like behavioral responses, serotonergic axonal sprouting and upregulated neurogenesis in the dentate gyrus, occluded further antidepressant-like action of fluoxetine [54], suggesting a ceiling effect in BDNF receptor-mediated fluoxetine-response. This evidence highlights that serotonergic and BDNF signaling in the hippocampus might influence each other in an auto-paracrine loop.

Other studies suggest that antidepressants may increase BDNF expression in cells other than neurons, albeit at lower levels. The incubation of rat primary cultured neurons with amitriptyline, at therapeutically relevant concentrations, did not affect BDNF expression [55]. This was in accordance with previous findings, showing monoamines presence is required for hippocampal and cortical neuronal cultures to show antidepressant-induced activation of BDNF promoters [56]. However, amitriptyline incubation, as well as clomipramine, fluvoxamine or duloxetine, but not cocaine, produced a rapid increase in *Bdnf* mRNA expression in astrocytes and microglia through an ERK-dependent pathway [55]. This evidence suggests a glial expression of BDNF, not dependent on the presence of monoamines that could contribute to the therapeutic effect of antidepressants. Furthermore, BDNF exerts a positive feedback effect on its transcription, through an autocrine/paracrine mechanism [14].

### 2.3. Epigenetic Regulation of the Bdnf Gene

The increased *Bdnf* mRNA variant expression induced by antidepressant treatment is mediated through epigenetic modifications, including DNA methylation and histone modifications [42,57]. DNA methylation can directly silence genes through the formation of methylcytosine in CpG dinucleotide-rich regions known as CpG islands, which subsequently perturbs gene transcription near transcriptional start sites [58].

Although recent work examining the link between *Bdnf* gene methylation and antidepressant medication in depressed patients did not provide consistent results [59,60,61,62,63], the variable-controlled environment of preclinical research showed decreased expression of the DNA demethylation machinery, able to target *Bdnf* transcripts, in mice exposed to unpredictable chronic mild stress [64]. Additionally, when DNA methylation was prevented by the systemic or hippocampal administration of DNMT1 inhibitor 5-azacytidine (decitabine), an antidepressant-like effect in the forced swim test was observed in rats, accompanied by decreased global DNA cytidine methylation and increased *Bdnf* levels in the hippocampus [65].

Apart from altering the degree of DNA methylation, antidepressants, including fluoxetine and escitalopram, modulate the phosphorylation of methyl-CpG binding protein 2 (MeCP2), a protein that decreases *Bdnf* transcription by occluding the *Bdnf* exon IV promoter activity [66,67]. Notably, knock-in mice with a non-functional MeCP2 phosphorylation site at Ser421 displayed depressive-like behavior and antidepressant resistance. When mice were exposed to chronic social defeat stress, they showed social avoidance, which was not rescued by chronic imipramine administration [68].

In addition, the ever-unfolding epigenetic modifications at *Bdnf* promoters revealed a functional role of histone modifications in antidepressant action. Early life stress and chronic restraint stress decreased histone H3 acetylation at *Bdnf* promoters, and downregulated expression of hippocampal BDNF, which were normalized by escitalopram treatment [67,69].

## 3. Lifestyle Interactions with Antidepressants in Rodent Models and Humans

Lifestyle involves a great diversity of factors such as nutrition, life events and the time that passes while we progress through our lives. All these factors have determinant effects on depression pharmacotherapy efficacy. Thus, although current antidepressants usually alleviate symptoms, delays in the onset, decrease, and lack of effects have been observed. Besides individual genetic and epigenetic propensities, the efficacy of antidepressants may therefore be, at least in part, explained by lifestyle interactions. Moreover, such lifestyle interactions induce changes in *Bdnf* gene expression regulated by epigenetic modifications such as DNA methylations and histone modifications.

### 3.1. Nutrition

A fundamental and well-studied aspect of lifestyle is nutrition, in which BDNF has an important role in the food intake control via reward learning [70]. A considerable effort has been made to understand the role of nutrition on antidepressant efficacy and disease management in addition to the MDD symptoms themselves [71,72,73,74]. Table 1 summarizes the clinical trial data addressing the impact of adjunctive nutrients on antidepressant therapies.

A deficiency in long-chain omega-3 fatty acids (omega-3 FA) is generally associated with mood disorders including major depressive disorder (MDD) and it also influences the treatment response. Dietary omega-3 FA themselves support *Bdnf* expression in the hippocampus of rats [75], through calmodulin kinase II and activated Akt [76], and increase *Bdnf* regulators such as CREB [77]. Supplementation with omega-3 FA increased serum BDNF levels and improved depressive symptoms in a randomized controlled trial with children and adolescents with depressive disorder [78]. Omega-3 FA such as Eicosapentaenoic acid and Docosahexaenoic acid used in combination with various antidepressants generally improved the antidepressant response [79,80,81,82,83,84,85,86,87]. Consistently, a randomized, double-blind, placebo-controlled study of a combination therapy with citalopram and omega-3 FA revealed a higher efficacy of antidepressant treatment when both components were administered together [88]. Moreover, in a randomized, double-blind, placebo-controlled dose-escalation study with the SSRI paroxetine, treated patients showed higher FA-chain length, unsaturation, and peroxidation. Negative relationships between FA and cortisol were associated with paroxetine nonresponse, which was also associated with low docosahexaenoic acid and low fatty fish intake [89]. In a 6-week double-blind, randomized, and placebo-controlled trial, therapy with Palmitoylethanolamide, a FA, and citalopram exerted a higher and faster antidepressant effect than in the absence of this FA [90]. Accordingly, in an animal model of FA insufficiency, rats fed without the omega-3 FA precursor alpha-linolenic acid and chronically treated with fluoxetine (FLX) exhibited greater climbing behavior, alpha2A adrenergic receptor mRNA expression, and lower 5-HT1A mRNA expression compared with rats fed with normal food [91]. However, omega-3 FA deficiency did not significantly reduce the effects of chronic FLX treatment on serotonin turnover or behavior in the forced swim test in female rats [92].

In line with an altered fatty acid metabolism, abnormalities in serum cholesterol levels in patients with mood disorders have been epidemiologically identified (reviewed in [93]). In rats fed with a cholesterol-enriched diet and subjected to chronic stress and anxiety paradigms, diazepam and clomipramine effects were not altered [94].

The relationship between caloric intake and the etiology of mood disorders has also gained attention in recent decades (reviewed in [95]). Intermittent fasting itself increased levels of BDNF and ameliorated anxiety and depressive-like behaviors in a type 2 diabetes mellitus model in rats [96]. However, differential effects were reported in case of MDD patients with mild to severe symptoms [97]. In mice, 9 h of fasting and imipramine treatment potentiated antidepressant-like effects and increased the ratio of phospho-CREB/CREB. These effects were partially reversed by treatment with the 5-HT2A/2C receptor agonist, (±)-1-(2, 5-dimethoxy-4-iodophenyl)-2-aminopropane hydrochloride. Furthermore, the effect of DOI hydrochloride was stronger compared to the non-fasting control group [98,99]. Moreover, mice fed with a high-fat diet, and exposed to social defeat stress showed a diminished antidepressant response to FLX [100].

The phosphocreatine pathway is vital for replenishing the energy donor ATP in the brain and is involved in MDD pathophysiology [101]. Subchronic administration of creatine alone showed antidepressant-like effects that were dependent on Akt activation and increased expression of BDNF in the hippocampus of mice [102]. In a randomized, double-blind placebo controlled trial an increase in creatine as adjunctive treatment to escitalopram enhanced the efficacy of the antidepressant in women [103]. In female adolescent patients with FLX-resistant depression, the Children’s Depression Rating Scale-Revised (CDRS-R) raw score decreased by 56% following adjunctive creatine treatment [104]. Similarly, only female rats supplemented with creatine showed a significant increase in the FLX antidepressant-like behavior compared to FLX alone [105].

Alterations in serotonergic neurotransmission have been considered one of the main causes of neuropsychiatric disorders including MDD during the 1960s [106]. However, more reliable evidence came from experiments using tryptophan depletion approaches [107]. A decrease in the essential amino acid tryptophan, the main metabolite in serotonin synthesis, may disturb serotonin synthesis [108]. In rats fed with a low tryptophan chow, fluvoxamine and fenfluramine diminished 5-HT release and, in rats fed with a high tryptophan diet, fenfluramine enhanced serotonin release [109]. Clinical data show an improvement in response to fluoxetine, chlorimipramine and phenelzine when adjunctive tryptophan to these antidepressants is used [110,111,112].

Methionine is another essential amino acid crucial to mental health. The methionine-derivative S-adenosyl methionine, cofactor in the single carbon cycle is involved in the synthesis of serotonin, noradrenalin and dopamine as well as in providing methyl donors for DNA- and histone methylation [113]. Clinical data demonstrated the efficacy of adjunctive S-adenosyl methionine use to improve antidepressant response [73,114,115], even in case of selective serotonin reuptake inhibitor (SSRI)-treatment resistant patients [116]. Moreover, N-acetylcysteine, a derivative of the amino acid cysteine, typically prescribed for bronchopulmonary disorders, improved antidepressant response when used as a treatment adjunctive for MDD [114]. However, the beneficial effects of S-adenosyl methionine and N-acetylcysteine and were not moderated by BDNF [117,118]. The dietary supplement N,N-dimethylglycine, a derivative of the amino acid glycine, acts at the glycine binding site of the NMDA receptor. Therefore, it enhances antidepressant-like effects of ketamine, but does not affect ketamine-induced anesthesia in mice [119]. Similarly, the dietary supplement betaine, a methyl derivative of glycine, increases the antidepressant-like effects but blocks the psychotomimetic effects of ketamine in mice [120].

Vitamins are also essential to our health. Evidence has suggested that folic acid, a water-soluble vitamin of the B complex group, can be used for enhancing antidepressant therapy in patients with deficient folic acid levels [121], and usually this deficiency is related to loss of responsiveness to conventional antidepressants [122,123,124]. Folic acid itself affected antidepressant-like behaviors and modulated BDNF in a chronic unpredictable mild stress rat model [125]. Clinical data show an improvement in the SSRI/SNRI antidepressant response in the presence of folic acid or its’ biological active form L-methylfolate [126,127,128,129,130]. However, this effect could not be demonstrated in the response of patients treated with other antidepressants, suggesting a role in serotonin and norepinephrine reuptake [131]. Furthermore, vitamin B12, like folate a member of the one carbon metabolism, improved the response to SSRI and tricyclic antidepressants in MDD-patients [132]. In a mouse model of combined chronic and acute stress, vitamin B12 increased stress resilience and restored TrkB levels [133]. However, no clear potentiation of antidepressant medication by B12 supplementation in combination with other vitamins on depressive symptomatology was reported in two clinical trials with patients treated with diverse antidepressant therapies [134,135]. Other vitamins including vitamin C and vitamin D3 also showed promising effects on the response to SSRIs fluoxetine and citalopram [136,137,138,139]. The effect of vitamin D3 in particular might be mediated by BDNF [140]. However, more research is needed in this direction.

Minerals are very important in several cellular processes. More than 300 enzymes and around 2000 transcription factors require zinc for their function [141,142]. The effect of zinc itself on depression has been broadly studied, and an interaction between zinc, BDNF and neuropeptides is emerging [143]. In mice fed with a zinc-deficient diet, BDNF levels and the antidepressant-like effects of both imipramine and escitalopram were reduced [144,145], with a reduction in the serum zinc levels and an increase in the corticosterone level [146]. Moreover, FLX evoked antidepressant-like effects and blocked the zinc restriction-induced changes in hippocampal p-CREB but not BDNF protein levels in rats fed with a zinc-deficient diet [147]. Clinical data show an improvement in the antidepressant response to fluoxetine, citalopram and imipramine in the presence of zinc as adjunctive compound [148,149,150,151]. Magnesium, the second most abundant intracellular cation [152], which in itself may have antidepressant properties [153], did not affect the efficacy of FLX in MDD-patients at any stage of treatment [154].

### 3.2. Environment

In addition to each person’s unique genetic heritage, the individual’s environment plays a crucial role in leaving a distinct epigenetic footprint that may predispose to or provide resilience against mental illnesses such as MDD. Studies on chromatin marks are thus necessary to understand the etiology and pathology of MDD [155]. Accordingly, changes in the epigenome have been related to antidepressant response as well. Altered DNA methylation on the promoter region of the serotonin transporter SLC6A4 has been associated with impaired antidepressant treatment response [156,157]. Furthermore, DNA methylation and related enzymes have been suggested to affect the therapeutic effects of tricyclic antidepressants, SSRIs and valproate [158,159,160]. Moreover, an association in the DNA methylation state from MDD patients with paroxetine response has been reported [161].

Great efforts have been made to find environmental predictors for better antidepressant responses. Studies on gene-environment interactions may not only help to understand the pathophysiology of MDD but could also provide predictive factors for personalized antidepressant therapy [162,163]. However, a number of studies have not found consistent associations and a more complex relationship between environment and treatment is likely [164,165,166,167]. Hence, individualized therapy is gaining importance, as the sum of genetic risk and environmental factors results in a differential and individual response to antidepressants [168,169].

Environmental conditions, including stressful or traumatic events are known risk factors to depression [170]. Accordingly, chronic stress alters the expression of neurotrophic factors including BDNF and could thereby inhibit signaling pathways that mediate antidepressant effects [171]. During stressful situations, hypothalamic corticotropin-releasing hormone (CRH) is released, which elevates cortisol levels. Cortisol is increased in patients suffering from MDD [172] and is thought to impair serotonin transmission [173]. In a study of CRH polymorphisms and stressful life events in patients with MDD, authors suggested an association between CRH ht1 haplotype, which is moderated by stressful events, and the antidepressant treatment outcomes [174]. In chronically stressed mice, treatment with ketamine increased antidepressant-like effects in stressed animals but had the opposite effect in unstressed controls that were housed in an enriched environment [175].

One of the most useful environmental predictors of antidepressant response in MDD are traumatic childhood experiences [176]. It has been reported that traumatized patients respond better to psychotherapy than to pharmacological treatments [177]. Moreover, individuals with traumatic childhood experiences showed generally lower treatment responses across various treatment approaches [178]. In addition, an association between higher levels of childhood abuse and depression severity has been reported in patients with a weak affinity to the serotonin transporter who received antidepressant therapy [177].

Adult mice exposed to early life stress or adult fluoxetine treatment did not show changes in hippocampal neurogenesis. Antidepressants only affected neurogenesis in adolescent that were not exposed to early life stress, suggesting that antidepressant action works in a distinct background of age and prior stress exposure [179]. In male pups exposed to the early life stress model of limited bedding, authors observed a positive correlation between stress and TrkB expression in the prefrontal cortex, and a negative association in the hippocampus [180]. Prenatal stress can also have lasting effects on treatment responses later in life. For instance, in rodents, stress in utero increased methylation at *Bdnf* promoters, which was associated with a reduced *Bdnf* transcription and increased depressive-like behaviors in tail suspension, forced swim, and sucrose preference tests [181]. In addition, prenatal stress altered stress coping in male and female juvenile rats and increased *Bdnf*-expression [182]. Intergenerational effects of early life stress on *Bdnf* promoter methylation have been reported in rats as well. However, it is unclear, how this influences their response to antidepressants [183].

Conversely, pleasant life events have positive effects on depression treatment. In rats, environmental enrichment improved behaviors associated with depression but not anxiety [184]. Environmental enrichment for several weeks consistently increases BDNF levels in rodents [185]. Treatment with the SSRI sertraline was more effective in diminishing depressive-like behavior in the presence of standard or environmental enrichment housing compared to social isolation conditions. Conversely, early social enrichment increased adult hippocampal BDNF levels in mice [184].

Climatic changes in the environment, such as weather, are a well-known mood influencer. Accordingly, diminished light exposure affects BDNF levels as well [186]. A change in the monthly sunshine duration exerted a significant effect on paroxetine response time, with a later onset of treatment response during shorter sunshine duration [187].

In addition, sport or physical activities are beneficial for depression treatment [188]. In line with this, a rat model of depression showed higher mRNA levels of *Bdnf*, and the presence of a running wheel had a positive effect on the treatment response with escitalopram [189]. In rats with colorectal cancer and depressive-like behavior, treatment with quercetin and exercise improved anti-tumor and antidepressant effects by suppressing inflammatory markers and upregulating the BDNF/TrkB/ß-Catenin axis [190]. Exercise improved depressive-like behaviors in ovariectomized rats and BDNF levels in hippocampus [191], increased the expression of 5-HT, BDNF and TrkB, and improved the mobility in the forced swim test in socially isolated rat pups [192]. Furthermore, exercise alleviated stress-induced social isolation via 5-HT1A receptor activation in rats [193]. Release of ß-hydroxybutyrate after prolonged exercise is associated with an increase in the activity of the *Bdnf* promoter by HDAC2 and HDAC3 in mice [194]. In addition, differential *Bdnf* methylation was associated with exercise in Vietnam veterans that suffered from posttraumatic stress disorder [195]. Taken together, exercise has reproducible effects on BDNF levels, which may, at least in part, mediate its’ antidepressant and mood elevating effects.

### 3.3. Aging

Although aging has received much attention recently, age-dependent expression of BDNF-TrkB and the mechanisms involved in depression require further investigation. Microarray data from the prefrontal cortex of healthy subjects confirmed an age-dependent BDNF downregulation and a positive correlation between BDNF and synaptic genes [196]. In addition, an age-dependent reduction in TrkB expression in the hippocampus is associated with altered spine morphology [197], a hallmark of depression [198]. Social defeat in adolescent, but not in adult rats, leads to an upregulation of BDNF-related immediate early genes [199]. Unpredictable chronic stress induced lower BDNF expression levels in aged hippocampus compared to young rats [200]. Moreover, enriched environment, which alleviates depressive-like behavior, increased *Bdnf* mRNA in the frontal cortex of in BDNF deficient mice independently of age, as well as in old wildtype mice [201]. Given the association between neurotrophin signaling, including BDNF and TrkB, with aging [202], an interplay between antidepressant response and age is likely. Many researchers have tried to optimize treatments for MDD patients of all ages; however, difficulties especially arise in elderly patients. This is partially due to a high variability in the efficacy and tolerability of antidepressants in the presence of age-related diseases, which require their own medicament therapies [203]. Hence, the association between the age of patients with MDD and the treatment efficacy is still not clear. While some studies reported an improved response [204], others found a lower antidepressant response in older patients [205]. In patients with MDD treated with paroxetine in a 6-week protocol, the late responders were significantly younger than the late or non-responders [179].

**Table 1 ijms-23-14445-t001:** Studies on the impact of dietary factors on antidepressant response.

Adjunctive Nutrient	Antidepressant	Outcome	Reference
Omega-3 PUFA 1000 mg/d	Sertraline	MADRS: Omega-3 PUFA < placebo	[81]
Ethyl-EPA 1 g/d	Fluoxetine 20 mg/d	HDRS: Ethyl-EPA < placebo	[79]
EPA 1, 2 or 4 g/d	Unspecified antidepressants	HDRS EPA < placebo (best 1 g/d)	[82]
EPA 2 g/d	Unspecified antidepressants	HDRS EPA < placebo	[84]
DHA 260 mg/d or 520 mg/d	Unspecified antidepressants	HDRS < after treatment resistance	[87]
EPA 1.8 g/d + DHA 0.4 g/d	Citalopram 20–40 mg/d	HDRS: Fatty acids < placebo	[88]
EPA 0.93 g/d + DHA 0.75 g/d	Sertraline 50 mg/d	BDI-II or HDRS: Fatty acids = placebo	[80]
EPA 1 g/d, DHA 1 g/d	Unspecified antidepressants	HDRS: EPA < DHA or placebo	[86]
EPA + DHA 3 g/d (EPA 0.6 g; DHA 2.4 g)	Unspecified antidepressants	HDRS-SF: EPA + DHA = placebo	[85]
EPA + DHA 9.6 g/d	Unspecified antidepressants	HDRS EPA + DHA < placebo	[83]
Palmitoylethanolamide 1200 mg/d	Citalopram 20 mg/d	HDRS: Palmitoylethanolamide < placebo	[90]
Creatine 4 mg/d	Fluoxetine 20–40 mg/d	CDRS-R: Creatine < placebo	[104]
Creatine 5 g/d	Escitalopram 20 mg/d	HDRS: creatine < placebo	[103]
Tryptophan 4 g/d	Fluoxetine 20 mg/d	HDRS: Tryptophan < placebo	[110]
DL-Tryptophan 0.1 g/kg body weight	Chlorimipramine 150 mg/d	Cronholm-Ottosson Depression Scale: n.s.	[111]
DL-Tryptophan 12, 15, 18 g depending on body weight	Phenelzine 60 mg/d	HDRS: DL-Tryptophan < placebo	[112]
SAMe 800 mg/d	Unspecified antidepressants	HDRS: < after treatment resistance	[206,207]
SAMe 1600 mg/d	SSRIs	HDRS: SAMe < placebo (in treatment resistance)	[116]
SAMe 200 mg/d, NAC 200 mg/d and folate 200 ug/d	SSRIs	HADS-A and CGI: better with SAMe, NAC and folate	[114]
L-Methylfolate 15 mg/d	SSRIs	QIDS-SR and CGI: better with L-Methylfolate in SSRI resistance	[129]
L-Methylfolate	SSRIs/SNRIs	CGI: L-Methylfolate < placebo	[130]
Folic acid 0.5 mg/d	Fluoxetine 20 mg/d	HDRS: Folic acid < placebo	[126]
Folic acid 10 mg/d	Fluoxetine 20 mg/d	HDRS: Folic acid < placebo	[127]
Folic acid 5 mg/d	Unspecified antidepressants	BDI-II: Folic acid = placebo	[131]
Folic acid 2 mg/d + Vitamin B12 0.5 mg/d + Vitamin B6 25 mg/d	Citalopram 20–40 mg/d	MADRS: Folic acid + B12 + B6 = placebo	[135]
Folic acid 400 ug/d + Vitamin B12 100 ug/d	Unspecified antidepressants	PHQ-9: n.s.	[134]
Vitamin B12 1 mg/d	SSRIs 20–40 mg/d	HDRS: with Vitamin B12 < placebo	[132]
	TCA 100–200 mg/d	HDRS: with Vitamin B12 < placebo	[132]
Vitamin C 1 g/d	Fluoxetine 10–20 mg/d	CDRS: with Vitamin C < placebo	[136]
	Citalopram 20 mg/d	HDRS: with Vitamin C = placebo	[138]
Vitamin D3 1500IU/d	Fluoxetine 20 mg/d	HDRS: with Vitamin D3 < placebo	[137]
Vitamin D3 300.000 U	Unspecified antidepressants	HDRS with Vitamin D3 < placebo (after 4 weeks)	[139]
Zinc 25 mg/d	Fluoxetine 20–60 mg/d	BDI: Zinc < placebo	[148]
	Citalopram 20–60 mg/d	BDI: Zinc < placebo	[148]
	Imipramine 100–200 mg/d	BDI, HDRS, CGI, MADRS: Zinc < placebo	[150,151]
	Unspecified antidepressants	HDRS, BDI: Zinc < placebo	[208]
Magnesium 120 mg/d	Fluoxetine 20–40 mg/d	HDRS: Magnesium = placebo	[154]

Abbreviations: BDI: Beck Depression Inventory; CDRS: Children’s Depression Rating Scale-Revised raw score; CGI: The Clinical Global Impression; DHA: Docosahexaenoic acid; EPA: Eicosapentaenoic acid; HDRS: Hamilton Rating Scale for Depression; MADRS: Montgomery–Åsberg Depression Rating Scale; NAC: N-acetylcysteine; SAMe: S-adenosyl methionine; n.s.: not significant; PHQ-9: Patient Health Questionnaire-9; QIDS-SR: Quick Inventory of Depressive Symptomatology. Placebo refers to treatment without corresponding adjunctive nutrient. Scale values lower in experimental cases than in placebo refer to a better outcome, while equal mean refers to not significance in effects.

## 4. Conclusions

The response to antidepressants is multifactorial, and the great variability in treatment response has increased the interest in developing personalized treatment strategies. However, the difficulty of managing or controlling lifestyle factors makes it challenging to design such studies, particularly in humans, and many underlying mechanisms still need to be elucidated. Nevertheless, there is mounting evidence that BDNF signaling can integrate diverse environmental and therapeutic cascades and holds promise as a pharmacological jack of all trades in the depression field.

## Figures and Tables

**Figure 1 ijms-23-14445-f001:**
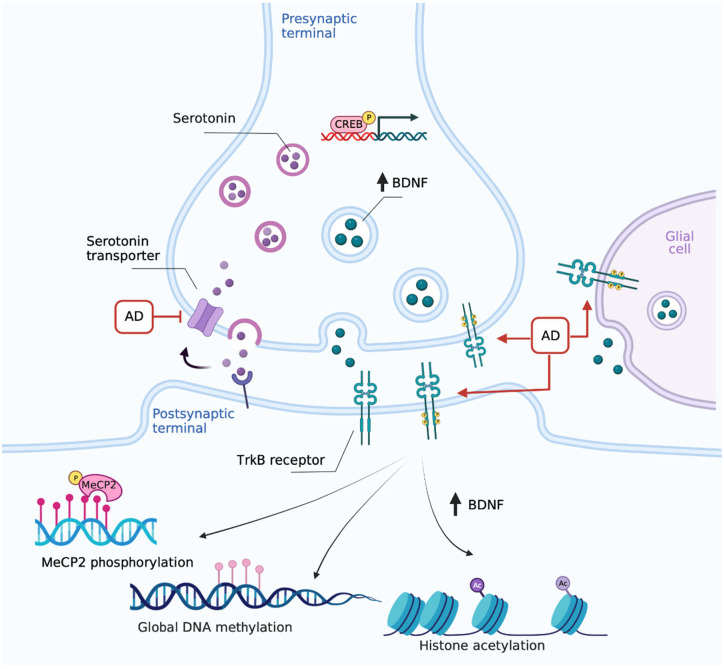
The paradigm shift in antidepressants’ mechanism of action: focus on TrkB. Chronic monoaminergic antidepressants (AD) increase hippocampal and cortical BDNF levels promoting autocrine/paracrine mechanisms, which include the direct interaction with TrkB and CREB-mediated upregulation in serotonergic terminals. Created with BioRender.com.

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
