# Peer review of "BDNF as a Mediator of Antidepressant Response: Recent Advances and Lifestyle Interactions"

_ijms, 2022, doi:10.3390/ijms232214445_

Round 1
Reviewer 1 Report
The review by Cubillos et al. focuses on the BDNF-TrkB signaling as a key transducer of antidepressant response. Authors also consider nutrition and environment as an important factors underlie antidepressant efficacy. This topic is timely and important but I have some considerations regarding the manuscript as follows:
1 1) Structurally the review looks disproportionate. Authors substantially focused on the BDNF and TrkB as antidepressant mediators and targets. At the same time, implication of BDNF and TrkB in the early life stress and physical activity covered much less. The effect of nutrition on the BDNF levels and signaling also largely uncovered. The section 4.3. need to be supported by examples of experimental data on age-dependent fluctuations in BDNF-TrkB expression. I believe that without expanding these parts, the review will lose its scientific value and be less interesting to potential readers.
2 2) Authors should keeping in mind that the besides neurotrophins (p. 1, lines 38-41) delay in therapeutic effect of antidepressants is depends on 5-HT receptors desensitization. It is well known fact and it is actual in light of new data on physical interaction between TrkB and 5-HT2A receptors (Ilchibaeva et al., 2022, PMID: 35954229).
3 3) Dysfunctional TrkB as etiological factor for depression supported not only by experimental data (p.6, lines 238-242) but also by the postmortem findings in MDD subjects (Dwivedi et al., 2003, PMID: 12912764; Pandey et al., 2008, PMID: 18611289; Dwivedy et al., 2009, PMID: 18930453; Banerjee et al., 2013, PMID: 24031163).
Minor: Throughout the text, the abbreviation of the Bdnf gene should be in italics.
Author Response
1) Structurally the review looks disproportionate. Authors substantially focused on the BDNF and TrkB as antidepressant mediators and targets. At the same time, implication of BDNF and TrkB in the early life stress and physical activity covered much less. The effect of nutrition on the BDNF levels and signaling also largely uncovered. The section 4.3. need to be supported by examples of experimental data on age-dependent fluctuations in BDNF-TrkB expression. I believe that without expanding these parts, the review will lose its scientific value and be less interesting to potential readers.
We appreciate the reviewer's constructive comments. In accordance, the first part of the manuscript has been substantially revised to focus on recent advances regarding antidepressants' mechanism of action. As such, paragraphs 1 and 3 were merged, while less relevant sections (i.e. “Distinct patterns of BDNF splicing variants…”) were deleted. In addition, the order of the second paragraph was revised, presenting antidepressant-induced effects on BDNF signaling in the order from brain regions to cell types, to intracellular events. Furthermore, we included further references regarding BDNF and its’ regulation by environmental factors and aging. We hope that the reviewer will find the manuscript more proportionate and clearer in its new form.
2) Authors should keeping in mind that the besides neurotrophins (p. 1, lines 38-41) delay in therapeutic effect of antidepressants is depends on 5-HT receptors desensitization. It is well known fact and it is actual in light of new data on physical interaction between TrkB and 5-HT2A receptors (Ilchibaeva et al., 2022, PMID: 35954229).
The text was amended according to the reviewer's indication (page 1, lines 20-25). The reference suggested has been also included in the text (page 3, lines 39-42).
3) Dysfunctional TrkB as etiological factor for depression supported not only by experimental data (p.6, lines 238-242) but also by the postmortem findings in MDD subjects (Dwivedi et al., 2003, PMID: 12912764; Pandey et al., 2008, PMID: 18611289; Dwivedy et al., 2009, PMID: 18930453; Banerjee et al., 2013, PMID: 24031163).
We are grateful for this valuable suggestion. The references indicated have been added (page 3, lines 18-21).
Minor: Throughout the text, the abbreviation of the Bdnf gene should be in italics.
We adjusted the formatting accordingly.
Reviewer 2 Report
Cubillos and coworkers have written essentially two different reviews, one on the role of BDNF in antidepressant response and another about lifestyle factors in the same response. The two parts are only relatively loosely linked together. The first part, BDNF in the antidepressant response, although well updated with the literature, really brings no new insight into a topic that has already been extensively reviewed in several recent reviews. I find the environment part more interesting and less covered in recent literature. My general comment is that the authors should discard the first BDNF part and focus on the environmental part (for which 2/3 of the references are already now) and include some aspects of the BDNF part as mechanisms explaining the role of the environment. Below, I have listed some aspects that the authors should take into consideration when submitting a revised version.
One general really annoying thing is that the reference numbers in the text do not accurately reflect the numbers in the reference list. References have apparently been added and/or removed from the reference list without updating the text. Some comments related to references may therefore simply reflect the fact that the paper referenced to is wrong.
The first part, BDNF in the antidepressant response.
1. Although as such well updated with the literature, really brings no new insight into a topic that has already been extensively reviewed in several recent reviews. For example, the very first paragraph is entitled “The paradigm shift in antidepressant mechanism of action”, however, the paragraph only lists well-accepted concepts of BDNF regulation. The authors should better explain what exactly the paradigm shift is. Interesingly, there may actually be a paradigm shift ongoing, related to the recent finding by Casarotto et al. that the antidepressant effects would be produced by their direct binding to TrkB and not through monoamines or glutamate receptors at all. This issue is mentioned later in the review, but not here in the context of paradigm shift. Figure 1 also does not really help the reader to understand how BDNF mediates the antidepressant effect.
2. Line 98. The authors cite Alloyz et al. for antidepressant-induced increase in monoamines increasing BDNF release, however, there is nothing about antidepressants, serotonin or norepinephrine in this paper. The authors should find a proper reference or edit the text to match the reference.
3. Lines 131-134: The references cited explain differential BDNF regulation in different cell types or tissues, but not really in different subcellular compartments. The authors probably here refer to the differential usage of the 3’UTR and polyA-signals, but these papers have not been cited.
4. Lines 1229-145. The authors do not do a good job at explaining the complex structure of BDNF gene. It would be helpful to emphasize that all the splice variants only code for a single proBDNF protein in exon 9. Further, the fact that the authors mix old and new nomenclature does not really help the issue either.
5. Subtitle 2.4. is BDNF effects may be cell-type specific, however, there is nothing about BDNF effects in this paragraph. The title would be better changed to BDNF expression …
Part 2 Nutrition. As stated above, I suggest that this would become the main part, BDNF part only as a mechanistic explanation and much shortened.
1. Table 1. This is very good, but might be improved. Currently, the first column is Antidepressant, but the interest really is in the second column, Adjuctive nutrient. Now things like folic acid or zinc are appearing in many places. If not too much trouble, I would recommend swapping these columns and indicating all the data where different antidepressant regulate zinc, or folic acid, together. I think it would be more informative.
2. Also, it is not immediately clear what in the Outcome column Vitamin<placebo actually means. After some research, the reader may understand that the oucome in the listed scale is lower by vitamin, indicating better response. However, intuitively a readed might consider response to vitamin weaker than to placebo. But why not indicating this more clearly, either in the table somehow, or at least in the legend. Also, are all these responses really significant and if the are not, which ones are? This should be indicated, too.
3. Line 316, what is “this antagonist”? Does it refer to DOI or something else? Please clarify.
4. The statement on lines 327-328 seems rather strong, given the hefty discussion on this topic recently. Moreover, Cowen has published papers reaching an essentially opposite conclusion later, see for example DOI 10.1002/wps.20229.
5. Lines 387-392. The references given here on the environmental predictors of antidepressant response are really about GWAS, genetic rather than environmental predictions. Here the authors could rather cite papers coming from the STAR*D study or Igor Branchi’s lab.
Author Response
Reviewer 2
Cubillos and coworkers have written essentially two different reviews, one on the role of BDNF in antidepressant response and another about lifestyle factors in the same response. The two parts are only relatively loosely linked together. The first part, BDNF in the antidepressant response, although well updated with the literature, really brings no new insight into a topic that has already been extensively reviewed in several recent reviews. I find the environment part more interesting and less covered in recent literature. My general comment is that the authors should discard the first BDNF part and focus on the environmental part (for which 2/3 of the references are already now) and include some aspects of the BDNF part as mechanisms explaining the role of the environment. Below, I have listed some aspects that the authors should take into consideration when submitting a revised version.
One general really annoying thing is that the reference numbers in the text do not accurately reflect the numbers in the reference list. References have apparently been added and/or removed from the reference list without updating the text. Some comments related to references may therefore simply reflect the fact that the paper referenced to is wrong.
We apologize to the reviewer. We have updated the text in our literature software.
Regarding the link between parts 1 and 2 of the review: We believe it is important to first introduce what is known regarding BDNF and antidepressant response (part 1) before exploring how lifestyle factors can impact on antidepressant effects and underlying BDNF-signaling. We hoped that the extensive revisions explained above (reviewer 1), including streamlining part 1 and expanding part 2, will improve the link between both parts and make the review more smooth to read.
- Although as such well updated with the literature, really brings no new insight into a topic that has already been extensively reviewed in several recent reviews. For example, the very first paragraph is entitled “The paradigm shift in antidepressant mechanism of action”, however, the paragraph only lists well-accepted concepts of BDNF regulation. The authors should better explain what exactly the paradigm shift is. Interesingly, there may actually be a paradigm shift ongoing, related to the recent finding by Casarotto et al. that the antidepressant effects would be produced by their direct binding to TrkB and not through monoamines or glutamate receptors at all. This issue is mentioned later in the review, but not here in the context of paradigm shift. Figure 1 also does not really help the reader to understand how BDNF mediates the antidepressant effect.
We appreciate the reviewer's suggestions. In accordance, paragraphs 1 and 3 were merged in order to highlight the recent advances in antidepressant regulation of BDNF signaling (i.e. direct binding to the TrkB receptor). We hope that the referee will find the paragraph well-supported now.
In addition, as figure 1 is intended to describe the integrated and updated mechanism of action of antidepressants, the mention to it was moved to page 3, line 47. In addition, the figure title was modified.
- Line 98. The authors cite Alloyz et al. for antidepressant-induced increase in monoamines increasing BDNF release, however, there is nothing about antidepressants, serotonin or norepinephrine in this paper. The authors should find a proper reference or edit the text to match the reference.
We thank the referee; the references have been modified according to the suggestion.
- Lines 131-134: The references cited explain differential BDNF regulation in different cell types or tissues, but not really in different subcellular compartments. The authors probably here refer to the differential usage of the 3’UTR and polyA-signals, but these papers have not been cited.
In accordance with the general indication to improve the balance of the two review sections, this part has been deleted.
- Lines 1229-145. The authors do not do a good job at explaining the complex structure of BDNF gene. It would be helpful to emphasize that all the splice variants only code for a single proBDNF protein in exon 9. Further, the fact that the authors mix old and new nomenclature does not really help the issue either.
As mentioned, following the overall suggestion to lighten the first section of the manuscript, this paragraph was simplified, providing more focus on more relevant details. Furthermore, we did our best to be more consistent with the nomenclature.
- Subtitle 2.4. is BDNF effects may be cell-type specific, however, there is nothing about BDNF effects in this paragraph. The title would be better changed to BDNF expression …
The subtitle was modified in “Multiple cell types contribute to increased BDNF following antidepressant treatment”
Part 2 Nutrition. As stated above, I suggest that this would become the main part, BDNF part only as a mechanistic explanation and much shortened.
We shortened part 1 and expanded part 2 (see additional explanations above).
- Table 1. This is very good, but might be improved. Currently, the first column is Antidepressant, but the interest really is in the second column, Adjuctive nutrient. Now things like folic acid or zinc are appearing in many places. If not too much trouble, I would recommend swapping these columns and indicating all the data where different antidepressant regulate zinc, or folic acid, together. I think it would be more informative.
The reviewer has made an excellent observation. We changed the table accordingly.
- Also, it is not immediately clear what in the Outcome column Vitamin<placebo actually means. After some research, the reader may understand that the oucome in the listed scale is lower by vitamin, indicating better response. However, intuitively a readed might consider response to vitamin weaker than to placebo. But why not indicating this more clearly, either in the table somehow, or at least in the legend. Also, are all these responses really significant and if the are not, which ones are? This should be indicated, too.
We can see how our abbreviations may be confusion. Hence, to be more clear, we added a short explanation in the legend.
- Line 316, what is “this antagonist”? Does it refer to DOI or something else? Please clarify.
Yes indeed, “DOI” refers to DOI hydrochloride.The sentence has been changed accordingly.
- The statement on lines 327-328 seems rather strong, given the hefty discussion on this topic recently. Moreover, Cowen has published papers reaching an essentially opposite conclusion later, see for example DOI 10.1002/wps.20229.
The sentence was updated accordingly.
- Lines 387-392. The references given here on the environmental predictors of antidepressant response are really about GWAS, genetic rather than environmental predictions. Here the authors could rather cite papers coming from the STAR*D study or Igor Branchi’s lab
Thank you for the valuable information. The sentence was completed accordingly and additional references were included.
Round 2
Reviewer 1 Report
The authors adequately responded to all comments. I have no concerns on manuscript.